Diagnostic inertia in dyslipidaemia: results of a preventative programme in Spain

Palazón-Bru Antonio 1 antonio.pb23@gmail.com
Sepehri Armina 1
Ramírez-Prado Dolores 1
Navarro-Cremades Felipe 1
Cortés Ernesto 2
Rizo-Baeza Mercedes 3
Gil-Guillén Vicente Francisco 1
1 Department of Clinical Medicine, Miguel Hernández University , San Juan de Alicante , Spain
2 Department of Pharmacology, Paediatrics and Organic Chemistry, Miguel Hernández University , San Juan de Alicante , Spain
3 Department of Nursing, University of Alicante , San Vicente del Raspeig , Spain
Foti Daniela
Electronic publication date: 2015 Jul 28
Publication date: 2015
Volume: 3
Electronic Location ID: e1109
Received 2015 Apr 21; Accepted 2015 Jun 26
Copyright: © 2015 Palazón-Bru et al.
Copyright year: 2015
Copyright holder: Palazón-Bru et al.
License: This is an open access article distributed under the terms of the Creative Commons Attribution License, which permits unrestricted use, distribution, reproduction and adaptation in any medium and for any purpose provided that it is properly attributed. For attribution, the original author(s), title, publication source (PeerJ) and either DOI or URL of the article must be cited.
License URL: https://creativecommons.org/licenses/by/4.0/

Keywords: Physicians, Dyslipidaemia, Diagnosis, Primary health care, Cardiovascular risk factors

Funding: Conselleria de Sanitat (Valencian Community) The Conselleria de Sanitat (Valencian Community) provided permission and funding for this study. This public organism subsidized and authorized this study exclusively to determine the situation in patients attending their health centre. The grant was used to contract a person (Antonio Fernández) to collect and computerize the data. The funders had no role in study design, data collection and analysis, decision to publish, or preparation of the manuscript.

==============================
Others have analysed the relationship between inadequate behaviour by healthcare professionals in the diagnosis of dyslipidaemia (diagnostic inertia) and the history of cardiovascular risk factors. However, since no study has assessed cardiovascular risk scores as associated factors, we carried out a study to quantify diagnostic inertia in dyslipidaemia and to determine if cardiovascular risk scores are associated with this inertia. In the Valencian Community (Spain), a preventive programme (cardiovascular, gynaecologic and vaccination) was started in 2003 inviting persons aged ≥40 years to undergo a health check-up at their health centre. This cross-sectional study examined persons with no known dyslipidaemia seen during the first six months of the programme (n = 16, 905) but whose total cholesterol (TC) was ≥5.17 mmol/L. Diagnostic inertia was defined as lack of follow-up to confirm/discard the dyslipidaemia diagnosis. Other variables included in the analysis were gender, history of cardiovascular risk factors/cardiovascular disease, counselling (diet/exercise), body mass index (BMI), age, blood pressure, fasting blood glucose and lipids. TC was grouped as ≥/<6.20 mmol/L. In patients without cardiovascular disease and <75/≤65 years (n = 15, 778/13, 597), the REGICOR (REgistre GIroní del COr)/SCORE (Systematic COronary Risk Evaluation) cardiovascular risk functions were used to classify risk (high/low). Inertia was quantified and the adjusted odds ratios calculated from multivariate models. In the overall sample, the rate of diagnostic inertia was 52% (95% CI [51.2–52.7]); associated factors were TC ≥ 6.20 mmol/L, high or “not measured” BMI, hypertension, smoking and higher values of fasting blood glucose, systolic blood pressure and TC. In the REGICOR sample, the rate of diagnostic inertia was 51.9% (95% CI [51.1–52.7]); associated factors were REGICOR high and high or “not measured” BMI. In the SCORE sample the rate of diagnostic inertia was 51.7% (95% CI [50.9–52.5]); associated factors were SCORE high and high or “not measured” BMI. Diagnostic inertia existed in over half the patients and was associated with a greater cardiovascular risk.

Introduction

Despite the great advances in medicine, coronary heart disease (CHD) and stroke have remained the main causes of death worldwide for over a decade (WHO, 2014). Mathematical models based on scoring systems have been used to determine the main risk factors for these diseases. These factors can be classified as non-modifiable (male gender and older age) and modifiable (altered lipid levels, diabetes mellitus, high blood pressure and smoking). These latter factors should be acted on via early detection and control, with the aim of reducing the incidence of cardiovascular diseases (Wilson et al., 1998; Conroy et al., 2003; Marrugat et al., 2007; WHO, 2007).

The detection of dyslipidaemia (altered lipid levels in blood) requires obtaining the blood concentration of lipids (mean of two fasting measurements) and determining whether that concentration is altered (NCEP & ATP III, 2002). In Spain in 2003, the total cholesterol (TC) concentration was used to diagnose dyslipidaemia (Villar Alvarez et al., 2003), with altered values being defined as TC ≥ 5.17 mmol/L (NCEP & ATP III, 2002; Villar Alvarez et al., 2003). This meant that, if a patient had a TC concentration above this threshold, the patient should undergo a second measurement to confirm or discard the diagnosis of dyslipidaemia (NCEP & ATP III, 2002; Villar Alvarez et al., 2003). If, after this second measurement, the physician then diagnoses dyslipidaemia, he or she should act according to the relevant guidelines. This action involves various possibilities, including dietary and hygiene measures or pharmacologic treatment (statins, fibrates and resins) (NCEP & ATP III, 2002; Villar Alvarez et al., 2003).

In 2010, diagnostic inertia was defined as a derivation of clinical inertia, as defined by Phillips in 2001 (Phillips et al., 2001; Gil-Guillén et al., 2010). Diagnostic inertia was defined as a situation in which a patient fulfilled the diagnostic criteria for a particular disorder but was not diagnosed by the respective physician as having this disorder (Gil-Guillén et al., 2010). A paper published in 2014 examined this concept in dyslipidaemia, considering the concentrations of TC and high-density lipoprotein cholesterol (HDL-C). This paper also undertook an exhaustive literature search detailing the main characteristics of those studies that had assessed failures in the diagnosis of dyslipidaemia, even though these studies had not in fact used the term diagnostic inertia to refer to these failures (Palazón-Bru et al., 2014).

Some of these authors evaluated the history of cardiovascular risk factors (CVRF) in relation to diagnostic inertia (Palazón-Bru et al., 2014), but none of them assessed the cardiovascular risk using a scoring system obtained through a multivariate model, considering the current status of the patients (current situation regarding CVRF; for example, current TC or HDL-C levels), rather than the already diagnosed CVRF. Therefore, we conducted a study calculating the cardiovascular risk in the patients using the following scoring systems: REGICOR (REgistre GIroní del COr) (calibration for Spanish persons of one of the scales of the Framingham study to evaluate morbidity and mortality due to coronary disease) and SCORE (Systematic COronary Risk Evaluation) (cardiovascular risk function in Europe, version TC/HDL-C) (Wilson et al., 1998; Conroy et al., 2003; Marrugat et al., 2007). We then examined the association of these cardiovascular risk scores in relation to the behaviour of the healthcare professionals. This type of association was examined by Sepehri et al. (2014) in their analysis of lack of advice to obese patients about losing weight. However, we have been unable to find any studies that analysed the situation as we proposed to do in this study. Accordingly, the results are innovative and highlight the need for measures to improve the diagnosis of dyslipidaemia.

Materials & Methods

Setting

The Valencian Community (Spain) had 4,518,126 inhabitants in 2004 (Instituto Nacional de Estadística, 2004). Primary healthcare is provided at health centres; coverage is universal and there is no cost to the patient. In this Community, a preventive activities programme following the recommendations of the Spanish guidelines was started at the end of 2003, aimed at the whole population aged 40 years or over (Pareja Bezares et al., 1999; Villar Alvarez et al., 2003; Robledo de Dios et al., 2003; Del Cura González, Arribas Mir & Coutado, 2003). Each person was invited, first by normal mail and then by telephone, to attend their health centre for a preventive study by medical and nursing personnel. The participants were given a report with the result of the examination and opportune recommendations, a copy of which was kept at the health centre. This programme included cardiovascular and gynaecologic screening and a vaccination campaign. The healthcare professionals who participated in this programme were dedicated solely to this job.

Study population

The study comprised all the patients who attended their health centres for the preventive activities programme. The main characteristics of these persons were: high prevalence of CVRF, older age, mostly women, and frequent attenders (Pedrera Carbonell et al., 2005).

Study design and participants

This was a cross-sectional study that analysed a sample of all individuals without known dyslipidaemia who were ≥40 years and who participated during the first six months in the preventive activities programme of the Valencian Community but who had an altered screening for TC (TC ≥ 5.17 mmol/L) (NCEP & ATP III, 2002; Villar Alvarez et al., 2003). Dyslipidaemia was defined according to the ICD-9-CM (272.x). Any patient who failed to fulfil these requirements was excluded from this study.

Variables and measurements

For this study we included all the cardiovascular data measured during the programme. Diagnostic inertia was the primary outcome variable. A patient was considered to experience diagnostic inertia if the healthcare professional failed to interpret correctly an altered TC screening result; that is the professional failed to start a follow-up programme of the patient to confirm or discard a diagnosis of dyslipidaemia.

The following secondary variables were considered: gender; personal history of diseases (hypertension, diabetes, CHD and stroke), personal history of smoking, counselling about diet and exercise (Yes or No), body mass index (BMI) (in kg/m2), age (in years), systolic and diastolic blood pressure (in mmHg), fasting blood glucose (FBG) (in mmol/L) and lipid profile (TC and HDL-C, in mmol/L).

The BMI, FBG, blood pressure and lipid profile were obtained using the standard methods. In some patients the BMI was not determined, in which case it was recorded as “not measured.” Data concerning the personal history of diseases, the smoking habit, the gender and the age were obtained at the patient interview and corroborated from the charts. Any counselling was recorded in the clinical history.

After obtaining all the data, the variables were grouped as follows (WHO, 1997): (1) BMI: “not measured”, low or normal (BMI < 25 kg/m2), overweight (BMI ≥ 25 kg/m2 and BMI < 30 kg/m2) and obese (BMI ≥ 30 kg/m2); (2) personal history of cardiovascular disease (CVD): having had CHD or stroke; and (3) TC values: TC ≥ 6.2 mmol/L and TC < 6.2 mmol/L (NCEP & ATP III, 2002; Villar Alvarez et al., 2003).

For each patient all the parameters in the preventive programme were recorded during one morning (systolic and diastolic blood pressure, FBG, BMI, TC and HDL-C). After collection, and on the same morning as the collection, the physician met the relevant patient and determined whether a future visit should or should not be arranged to confirm or discard a diagnosis of dyslipidaemia. In addition, at the same time the patient could receive personalized advice about exercise or diet.

After the collection and digitalization of the data and grouping the variables, the REGICOR cardiovascular risk function was ascertained in those patients for whom it was applicable. These patients were then classified in risk groups: high (≥20%) and low (<20%). The REGICOR scale estimates the 10-year probability of CHD in patients free of CVD between the ages of 30 and 74 years (Marrugat et al., 2007). CVD predictive variables: gender, age, lipid profile (TC and HDL-C), systolic and diastolic blood pressure, diabetes and smoking.

The same procedure was followed in those patients for whom the SCORE was applicable (version TC/HDL-C). The risk groups were (Conroy et al., 2003): high (≥5%) and low (<5%). The SCORE estimates the 10-year risk of cardiovascular death in patients aged 40–65 years with no previous CVD. The variables in this model are: age, gender, TC, HDL-C, systolic blood pressure, and smoking.

Sample size

The overall sample comprised 16,905 patients with no personal history of dyslipidaemia and with TC ≥ 5.17 mmol/L. Of these, 15,778 fulfilled the criteria for REGICOR evaluation and 13,597 for SCORE. Thus, using a significance of 5% and a maximum expected proportion (p = q = 50%), the expected error in the estimation of the proportion of diagnostic inertia was 0.75% in the overall sample, 0.78% in the REGICOR patients, and 0.84% in the SCORE patients.

Statistical analysis

The descriptive analysis was performed using the standard methodology in health sciences (frequencies, percentages, means and standard deviations). Multivariate logistic regression models were used to estimate the adjusted odds ratio (OR) in order to analyse the association between diagnostic inertia and the study variables. For the overall sample, the ORs were adjusted for gender; personal history of diseases and smoking, BMI group, counselling, TC group, and age as a quantitative variable. Another model was performed with the total sample using the current status of the CVRF (systolic blood pressure, TC and FBG), instead of the personal history of hypertension or diabetes mellitus, and TC ≥ 6.20 mmol/L. For the REGICOR sample, the ORs were adjusted for REGICOR risk group, BMI group, and counselling. For the SCORE sample, the ORs were adjusted for SCORE risk group, personal history of diabetes, BMI group, and counselling. The rest of the variables in all the models were not taken into account due to collinearity issues. The prognostic probabilities of inertia were also studied in multivariate models to produce charts in order to aid interpretation of the results. The goodness-of-fit of the models was tested by the likelihood ratio test. Furthermore, the unadjusted ORs were calculated for all the secondary variables. All the analyses were done with a level of significance of 5%, and the associated confidence interval (CI) of each parameter was calculated. All the analyses were done with IBM SPSS Statistics 19.

Ethical considerations

This study was approved by the Conselleria de Sanitat-Miguel Hernández University institutional review committee (Valencian Community) with reference number AVS-UV1.07X. This institution did not participate in data collection, analysis or interpretation, or in the decision to approve or disapprove publication of the final manuscript. In addition, the data were anonymized and encrypted, satisfying the data protection law.

As our study was population-based and non-interventional, using data from medical records, no informed consent was required. The institutional review committee approved this procedure and ensured that information access was completely restricted. In addition, its use was in line with current legislation.

Results

Total sample

Table 1 summarizes the information concerning the overall sample (n = 16, 905). Most of those who participated in the study were women (60.6%), there was a high prevalence of CVRF (hypertension, 17.3%; diabetes, 3.5%; smoking, 22.1%), almost 4% of CVD, and the vast majority of the patients had TC concentrations below 6.20 mmol/L (69.5%). Table 1 also shows the unadjusted ORs for all the secondary variables.

Table 1 Analysis of diagnostic inertia for dyslipidaemia at primary health care centres in the Valencian Community (Spain): 2003–2004 data.

ORs were adjusted for gender; personal history of hypertension, diabetes, smoking and CVD; BMI groups, counselling (diet and exercise), total cholesterol values (as a categorical variable) and age. Systolic and diastolic blood pressure, total and HDL cholesterol were not included in the multivariate model due to collinearity with the personal history of hypertension and TC ≥ 6.2 mmol/L. A second model was performed (current status of the cardiovascular risk factors) replacing TC ≥ 6.20 mmol/L, personal history of hypertension and diabetes by TC (in mmol/L), systolic blood pressure and fasting blood pressure. In this last model, diastolic blood pressure and HDL cholesterol were not included due to collinearity with systolic blood pressure and TC. Goodness-of-fit of the models: (1) Personal history of cardiovascular risk factors: X2 = 4, 750.1 P < 0.001; (2) Current status of the cardiovascular risk factors: X2 = 4, 834.1 P < 0.001.

Variable	Total	Inertia	Unadj. OR	P	Adj. OR	P	
	16,905	8,783 (52.0%)	(95% CI)		(95% CI)		
	n(%)/x±s	n(%)/x±sd					
Age (Years)	54.7±9.9	54.8±9.9	1.00(1.00, 1.01)	0.12	1.00(0.99, 1.00)	0.13	
REGICOR (probability of event)b	5.8±3.6	6.5±4.0	1.12(1.11, 1.14)	<0.001	N/M	N/M	
REGICOR risk groups:							
≥20%	128(0.8)	112(87.5)	6.57(3.89, 11.10)	<0.001	N/M	N/M	
<20%a,b	15,650(99.2)	8,074(51.6)					
SCORE (probability of event)b	1.3±1.9	1.6±2.3	1.22(1.19, 1.25)	<0.001	N/M	N/M	
SCORE risk groups:							
≥5%	551(4.1)	414(75.1)	2.94(2.41, 3.57)	<0.001	N/M	N/M	
<5%a,b	13,046(95.9)	6,616(50.7)					
Gender:							
Male	6,664(39.4)	3,679(55.2)	1.24(1.17, 1.32)	<0.001	0.97(0.91, 1.05)	0.49	
Femalea	10,241(60.6)	5,104(49.8)					
Personal history of hypertension:							
Yes	2,923(17.3)	2,087(71.4)	2.72(2.49, 2.96)	<0.001	4.27(3.85, 4.73)	<0.001	
Noa	13,982(82.7)	6,696(47.9)					
Personal history of diabetes:							
Yes	592(3.5)	351(59.3)	1.36(1.15, 1.61)	<0.001	1.16(0.96, 1.41)	0.13	
Noa	16,313(96.5)	8,432(51.7)					
Personal history of smoking:							
Yes	3,739(22.1)	2,779(74.3)	3.45(3.18, 3.74)	<0.001	4.94(4.50, 5.42)	<0.001	
Noa	13,166(77.9)	6,004(45.6)					
Personal history of CVD:							
Yes	639(3.8)	341(53.4)	1.06(0.91, 1.24)	0.47	0.86(0.71, 1.03)	0.10	
Noa	16,266(96.2)	8,442(51.9)					
BMI groups (kg/m2):							
<25a	4,157(24.6)	2,038(49.0)					
25–30	7,355(43.5)	3,803(51.7)	1.11(1.03, 1.20)	0.01	1.08(0.99, 1.18)	0.09	
≥30	4,357(25.8)	2,360(54.2)	1.23(1.13, 1.34)	<0.001	1.10(1.00, 1.22)	0.06	
Not measured	1,036(6.1)	582(56.2)	1.33(1.16, 1.53)	<0.001	1.26(1.07, 1.48)	0.01	
Diet counselling:							
Yes	14,407(85.2)	7,542(52.3)	1.11(1.02, 1.21)	0.02	1.02(0.91, 1.15)	0.72	
Noa	2,498(14.8)	1,241(49.7)					
Exercise counselling:							
Yes	14,369(85.0)	7,476(52.0)	1.02(0.93, 1.11)	0.71	0.99(0.88, 1.12)	0.86	
Noa	2,536(15.0)	1,307(51.5)					
Total cholesterol values (mmol/L):							
≥6.2	5,158(30.5)	4,189(81.2)	6.73(6.22, 7.29)	<0.001	8.33(7.63, 9.09)	<0.001	
<6.2a	11,747(69.5)	4,594(39.1)					
Systolic blood pressure (mmHg)c	128.9±17.3	130.5±17.9	1.01(1.01, 1.01)	<0.001	1.01(1.01, 1.01)	<0.001	
Diastolic blood pressure (mmHg)	78.4±10.4	79.2±10.7	1.02(1.01, 1.02)	<0.001	N/M	N/M	
Total cholesterol (mmol/L)c	6.0±0.7	6.2±0.8	5.90(5.50, 6.32)	<0.001	6.58(6.10, 7.09)	<0.001	
HDL cholesterol (mmol/L)	1.5±1.2	1.7±1.1	0.98(0.94, 1.02)	0.41	N/M	N/M	
Fasting blood glucose (mmol/L)c	5.5±1.4	5.5±1.4	1.09(1.06, 1.11)	<0.001	1.05(1.02, 1.08)	<0.001	
Notes.

Abbreviations

Adj. OR adjusted odds ratio

Unadj. OR unadjusted odds ratio

CI confidence interval

REGICOR REgistre GIroní del COr

SCORE Systematic COronary Risk Evaluation

HDL high density lipoprotein

CVD cardiovascular disease

BMI body mass index

N/M not in the models

a Reference.

b Only when the scoring system was applicable.

c Adjusted with the current status of the cardiovascular risk factors (systolic blood pressure, total cholesterol and fasting blood glucose).

d Prevalence of inertia.

The magnitude of inertia was 52.0% (95% CI [51.2–52.7]). Significantly associated factors (p < 0.05) in the multivariate models were: hypertension, smoking, higher BMI (overweight and obesity) or “not measured” BMI, TC ≥ 6.2 mmol/L and higher values of the control parameters of the CVRF (systolic blood pressure, FBG and TC). These results were similar to the unadjusted ORs (Table 1).

REGICOR sample

In the REGICOR sample (n = 15, 778), the proportion of inertia was 51.9% (95% CI [51.1–52.7]). Significantly associated factors were: a high REGICOR (≥20%) (OR = 6.49, 95% CI [3.84–10.98], p < 0.001), obesity or “not measured” BMI (BMI < 25 kg/m2 → OR = 1; BMI 25–29.9 kg/m2 → OR = 1.11, 95% CI [1.03–1.20], p = 0.08; BMI ≥ 30 kg/m2 → OR = 1.21, 95% CI [1.11–1.33], p < 0.001; BMI “not measured” → OR = 1.32, 95% CI [1.15–1.52], p < 0.001), and counselling about diet (OR = 1.13, 95% CI [1.01–1.25], p = 0.03).

Figure 1A shows a Cartesian chart of the risk groups according to the REGICOR on the X axis and prognostic probability of inertia on the Y axis. The chart shows that persons with a high risk had a greater probability of inertia.

Figure 1 Predicted probability of diagnostic inertia for dyslipidaemia for primary cardiovascular prevention patients.

Abbreviations: REGICOR, REgistre GIroní del COr; SCORE, Systematic COronary Risk Evaluation.

SCORE sample

In the analysis done for the SCORE sample (n = 13, 597) the proportion of inertia was 51.7% (95% CI [50.9–52.5]). Significantly associated factors were: a high SCORE (≥5%) (OR = 2.85, 95% CI [2.34–3.47], p < 0.001), diabetes (OR = 1.27, 95% CI [1.02–1.59]; p = 0.04), a higher BMI (overweight and obesity) or “not measured” BMI (BMI < 25 kg/m2 → OR = 1; BMI 25–29.9 kg/m2 → OR = 1.13, 95% CI [1.04–1.23], p = 0.01; BMI ≥ 30 kg/m2 → OR = 1.19, 95% CI [1.08–1.31], p < 0.001; BMI “not measured” → OR = 1.31, 95% CI [1.12–1.53], p < 0.001), and counselling about diet (OR = 1.15, 95% CI [1.03–1.29], p = 0.01).

Fig. 1B shows a Cartesian chart of the risk groups according to the SCORE on the X axis and prognostic probability of inertia on the Y axis. The chart shows that persons with a greater cardiovascular risk experienced more inertia.

Discussion

This study shows that a greater cardiovascular risk is related to experiencing diagnostic inertia when the physician fails in the interpretation of altered TC levels in patients who have no personal history of dyslipidaemia. This greater cardiovascular risk thus leads to an association between cardiovascular risk factors and diagnostic inertia. These risk factors were hypertension, higher systolic blood pressure, smoking, a high BMI (overweight and obesity) or “not-measured” BMI, higher TC concentrations, a high probability of developing a lethal or nonlethal CVD, a personal history of diabetes and higher values of FBG. In addition, dietary advice was also associated with diagnostic inertia.

Most previous studies differ from ours in these associations (Palazón-Bru et al., 2014), as only one paper found an association between the presence of cardiovascular risk factors and lack of a correct diagnosis of dyslipidaemia when there existed alterations in the levels of HDL-C (Palazón-Bru et al., 2014). On the other hand, the study by Sepehri et al. (2014) did, however, find an association between having a high REGICOR and experiencing clinical inertia for the treatment of obesity by personalized counselling to lose weight.

These results are of concern, as these patients have a very high likelihood of suffering CVD if no preventive measures are taken. Controlling blood pressure, the lipid profile, body weight, diabetes and ceasing to smoke are the main measures. Also of note was the greater inertia in persons offered personalized advice about adequate nutrition. This may be because healthcare professionals (physicians and nurses) prefer to give dietary advice rather than measure the TC, because the concentration of TC will decrease if the patient follows this advice. Furthermore, these results were seen during a cardiovascular preventive activities programme, in which the healthcare professionals had to either confirm or discard the diagnosis of dyslipidaemia. We found that half of the patients did not undergo a second measurement of TC, even though many were diagnosed with other CVRF and they should have had their lipid profile controlled to avoid CVD. This may be because most of these patients had multiple disorders and were polymedicated (Pedrera Carbonell et al., 2005), so that the healthcare professional decided not to order a dyslipidaemia confirmatory test at that time, as this would only increase the therapeutic complexity of the patient. Finally, a factor that we thought might have influenced the decision to monitor the patient was that the healthcare professional would accept borderline TC figures (TC < 6.20 mmol/L) as normal (NCEP & ATP III, 2002; Villar Alvarez et al., 2003). Our results, though, unexpectedly showed that greater inertia was committed in persons with non-borderline TC figures. Given that the aim of the study was restricted to determining factors associated with this inertia, a qualitative study should be undertaken to attempt to determine the reasons why the healthcare professionals fail to adhere to the clinical practice guidelines when diagnosing dyslipidaemia. This would enable us to corroborate the theories put forward (dietary advice instead of a second drug; patients with multiple diseases or on multiple drugs for whom the professional does not wish to increase the therapeutic complexity) and thus provide further understanding that would aid the healthcare professional when diagnosing dyslipidaemia, and which would hopefully result in an earlier diagnosis of the disorder.

The results of this study suggest the need to design measures aimed at reducing this diagnostic inertia. Worryingly, this diagnostic inertia was associated with a greater cardiovascular risk, both when assessed from a history of CVRF (or using the values of the control parameters) and when using a scoring system. A possible solution to this problem might be the inclusion of training courses in the health centres to remind the healthcare providers of the cut points for the diagnosis of the various CVRF. In addition, as suggested by others (Palazón-Bru et al., 2014; Sepehri et al., 2014), in order to reduce inertia it might be beneficial to integrate some sort of alarm in the computerized electronic records systems in the event that a TC measurement is abnormal. This would help the healthcare professional see that a diagnosis of dyslipidaemia should be confirmed in a patient who has an altered screening for TC. The end result of this could be a reduction in the incidence of CHD.

Limitations and strengths of the study

The main strength of this study is that it is the first to examine the association between diagnostic inertia in dyslipidaemia and the cardiovascular risk, as measured with probabilistic functions (REGICOR and SCORE). The results are therefore innovative. In addition, the large sample size and the participation of all the primary care physicians and nurses in the Valencian Community provide external validity to our conclusions.

Concerning selection bias, it should be recalled that we are evaluating data from a cardiovascular preventive programme, in which all the healthcare professionals had been instructed to undertake cardiovascular preventive activities with all their patients. This begs the reflection about what the situation would actually be in daily clinical practice; i.e., when the healthcare professionals follow their own criteria for ordering screening tests for the diagnosis of dyslipidaemia. This remains to be examined in future studies, which will then enable comparison with the present results. Another limitation concerns the fact that we were unable to include certain variables that could affect the outcome, such as psychosocial factors of both the healthcare professionals and the patients. This remains for future studies, once the above-mentioned qualitative study has been undertaken and the results analysed.

As regards measurement bias, the teams taking the measurements were requested to use reliable devices and to undertake the clinical interview correctly. Nevertheless, only TC was assessed as a lipid parameter to diagnose dyslipidaemia. The reason for not using low-density lipoprotein cholesterol, HDL-C or triglycerides is because in the year the preventive programme was carried out the Spanish clinical guidelines only advised the use of this parameter (Villar Alvarez et al., 2003). Finally, even though the data are from 2003–2004, we should nevertheless consider that the problem of diagnostic inertia (though it was not yet referred to as such) has been the subject of study since the 1980s. Even nowadays it still exists, despite newer guidelines and changes in the diagnostic cut points (Palazón-Bru et al., 2014). In other words, the results of this study, whilst not very recent, can still indicate the need for diagnostic inertia to be considered at the current time.

Conclusions

Diagnostic inertia occurred in over half the patients with no personal history of dyslipidaemia but who had an altered screening test and who attended a preventive activities programme. Worryingly, the healthcare professionals committed greater inertia in the patients who had a higher likelihood of CVD.

Given the relevance of the problem found, measures are needed to reduce the magnitude of inertia, particularly in the case of patients with a high cardiovascular risk.

The authors thank Ian Johnstone for help with the English language version of the text.

Additional Information and Declarations

Competing Interests

Author Contributions

Human Ethics

Data Deposition

Antonio Palazon-Bru is as an Academic Editor for PeerJ.

Antonio Palazón-Bru conceived and designed the experiments, performed the experiments, analyzed the data, wrote the paper, prepared figures and/or tables.

Armina Sepehri, Dolores Ramírez-Prado, Felipe Navarro-Cremades, Ernesto Cortés and Mercedes Rizo-Baeza conceived and designed the experiments, reviewed drafts of the paper.

Vicente Francisco Gil-Guillén conceived and designed the experiments, performed the experiments, reviewed drafts of the paper.

The following information was supplied relating to ethical approvals (i.e., approving body and any reference numbers):

This study is based on an institutional agreement between the Conselleria de Sanitat and Miguel Hernández University, Elche (reference number: AVS-UV1.07X) authorized by an Institutional Review Board (Universidad Miguel Hernández de Elche-Conselleria de Sanitat de la Generalitat Valenciana). The data were therefore analyzed in compliance with current legislation on medical ethics. Additionally, the data were anonymized and encrypted, in accordance with the data protection law.

This population-based, non-interventional study (data from the Valencian Community) used data from medical records and informed consent was not required for included patients. The institutional agreement approved this consent procedure and ensured that information access was restricted, it did not compromise the interests or welfare of any patient, it minimized the risk of injury and its use was in line with current legislation.

The following information was supplied regarding the deposition of related data:

The institutional review committee approved this procedure and ensured that information access was completely restricted; therefore, the database is not publicly available.

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
