# Peer review of "Diagnostic inertia in dyslipidaemia: results of a preventative programme in Spain"

_PeerJ, doi:10.7717/peerj.1109_

## Round 0.1 · original submission · Major Revisions

A common issue of both reviewers is that factors contributing to the complexity of diagnostic inertia in dyslipidemia should be better discussed.

Reviewer 1 ·

Basic reporting

This is an interesting paper which provides interesting information about cardiovascular risk and diagnostic inertia. The study methodology has clear limitations which are acknowledged.

Experimental design

In the Methods section (variables and measurements paragraph), I think you need to specify the length of follow-up.
For better interpretation of the multivariate analysis indicating that higher BMI is associated to greater probability of inertia, a precise definition of groups included in “higher BMI” group should be provided. Similar issues arise in all Tables.
In all Tables categories of BMI are overlapped (>=30 and 25-30).

Validity of the findings

The authors should emphasize factors contributing to diagnostic inertia. In particular, possible reasons because patients with abnormal screening results do not undergo a further investigation by physicians, should be more thoroughly discussed.

·

Basic reporting

No Comments

Experimental design

No Comments

Validity of the findings

No Comments

Additional comments

I think that your title is slightly inappropriate – please remove “effects of”…for better understanding, something as follows would be more suitable: Diagnostic inertia in participants with dyslipidaemia: results of a preventative programme in Spain

Line 39: what is meant by “…paying more attention to the current status of their patients…”?

Line 116 – should this not be ‘quantitative’?

Line 141 – please include the prevalence in brackets after women and for normal TC; and prevalence of CVRF

Tables 1-3: please include a footnote on the prevalence in brackets for the inertia columns
For BMI, please show p-value for each sub-category e.g. p-value comparing total vs. inertia for BMI >=30; p-value comparing total vs. inertia for BMI 25-30, etc.

Please move age to the top of your tables, above gender

Your reference results/values should preferably be presented first.

Tables 2 and 3 – there is no explanation for superscript ‘c’

There is much in the current Tables 2 and 3 that is similar to Table 1 and thus I believe that these tables do not add value.
I suggest that you include the mean systolic and diastolic BPs, and TC and HDL-C values in Table 1.
Table 2 would then simply include the mean and raised values for both REGICOR and SCORE with no Table 3. If there are any differences compared to Table 1 that you would like to emphasise, you may then include this in the text.

How common/widespread was treatment for hypercholesterolaemia in 2003-2004 in Spain – were statins easily available at the primary healthcare centres? What were the Spanish guidelines for treatment of hyperlipidaemia e.g. was a total cardiovascular risk approach used or was drug treatment simply recommended for all TC >=6.2 mmol/l, more importantly was dietary modification for 3-6 months the first recommendation for cholesterol lowering or was drug treatment to be initiated immediately? (Lines 174-176) If dietary change was the first line of treatment, then how would doing a 2nd test at this stage change management or be beneficial? Were these patients asked to return for follow-up? These factors would influence the healthcare provider’s behaviour/inertia to treatment and need to be discussed in detail.

What were the other risk factors screened for in the prevention programme? Lines 95-96 – did the physician only meet patients with abnormal TC results? Apart from hypercholesterolaemia, did the physicians impart a diagnoses of hypertension or diabetes to the patients? How many patients with newly diagnosed diabetes/hypertension were also newly diagnosed with hypercholesterolaemia? Perhaps a focus on the management of the former 2 conditions may have contributed to a lesser focus on raised TC and may partly explain treatment inertia in those with higher risk scores.

Did the prevention programme lead to overburdening of the primary healthcare facilities or were there more staff available/employed for the prevention programme? If not, then there may have been resentment towards the implementation of the screening/prevention programme and this could have led to treatment inertia.
Treatment inertia is a complex issue which the authors have not comprehensively addressed in their paper. While I acknowledge that you do not present psychosocial data, treatment inertia nonetheless has predominantly psychosocial determinants and your data need to be discussed in this context.

Lines 190 -192 – do you mean healthcare providers instead of participants?

---

## Round 0.2 · Minor Revisions

Although the authors improved their revised manuscript in the light of the reviewers' suggestions, I strongly recommend to address the remaining, constructive comments of reviewer 2.

Reviewer 1 ·

Basic reporting

Most of the methodological concerns raised in my review have been satisfactorily addressed in the revised manuscript.

Experimental design

no comments

Validity of the findings

no comments

·

Basic reporting

No Comments

Experimental design

No Comments

Validity of the findings

No Comments

Additional comments

The authors have attempted to address the reviewer’s comments. However, not all have been addressed satisfactorily.

Line 17 – please remove ‘figures’

Line 34 – please use the acronym HDL-C instead of HDL-c

Lines 39-42 – I still don’t follow the rationale about paying more attention to current medical conditions instead of known CVRF. Clarity is needed, please. Do you mean that healthcare professionals neglect the long-term management of CVD risk factors such as diabetes, hypertension, etc that are already present/known in their patients?
How would this be pertinent to this paper, please, if you have emphasised that this is a screening/diagnostic programme and not a treatment programme?

Line 73 – ‘Cross-sectional study’ – this needs to be a full sentence, please, preferably something as follows: This was a cross-sectional study that analysed all individuals without known dyslipidaemia who were ≥40 years of age and attended the first six months of the preventive activities….

Line 124 - As mentioned previously, the authors are incorrect in referring to n (%) as qualitative data. These are quantitative. Please amend accordingly.

Line 197 – please improve the sentence construction and meaning to avoid ambiguity ‘…as if the patient…:

As mentioned previously, the health professionals’ behaviours regarding screening need to be contextualised i.e. if treatment was not widely available then making a diagnosis of dyslipidaemia would be purely an academic exercise because it would not change the patient’s management. In such a scenario many healthcare professionals may not see a need to confirm a diagnosis of hyperlipidaemia. Thus, I strongly recommend, for greater clarity, that you please discuss/include whether the use of statins/anti-lipid agents in individuals diagnosed with dyslipidaemia was widespread or not in Spain in 2003/04.

Line 100 – it is not mentioned here but in the response to reviewers’ comments, the authors state that diabetes was also screened for. If this is so, then please include in the manuscript.

Table 1: As requested previously, I suggest that you present your reference data first, as is usual practice e.g. for the BMI category, present the data in the following order: <25 (reference), 25-29.9, >=30, not measured.

As requested previously, I strongly recommend calculating the p-value for each sub-category of BMI, particularly in view of the large number that did not have measured BMI.

I suggest that you present both the unadjusted data (currently not included) as well as the adjusted data for diagnostic inertia in your tables (can be in separate tables). Also, for the logistic regression analyses that are not related to the risk scores, please include blood pressure either as a continuous or categorical (hypertension) variable. The latter would include known hypertension. Also include HDL-C in the model, please, if it’s not highly correlated with TC levels. And glucose values if tested.

---

## Round 0.3 · accepted · Accept

The authors have satisfactorily responded to the concerns raised by the reviewer, and improved the quality of their manuscript.